# Theoretical Search for Gravitational Bound States of Tachyons

Charles Schwartz 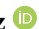

Department of Physics, University of California, Berkeley, CA 94720, USA; schwartz@physics.berkeley.edu

**Abstract:** The mission here is to see if we can find bound states for tachyons in some gravitational environment. That could provide an explanation for the phenomena called Dark Matter. Starting with the standard Schwarzschild metric in General Relativity, which is for a static and spherically symmetric source, it appears unlikely that such localized orbits exist. In this work, the usual assumption of isotropic pressure is replaced by a model that has different pressures in the radial and angular directions. This should be relevant to the study of neutrinos, especially if they are tachyons, in cosmological models. We do find an arrangement that allows bound orbits for tachyons in a galaxy. This is a qualitative breakthrough. Then we go on to estimate the numbers involved and find that we do have a fair quantitative fit to the experimental data on the Galaxy Rotation Curve. Additionally we are led to look in the neighborhood of a Black Hole and there we find novel orbits for tachyons.

**Keywords:** general relativity; Schwarzschild metric; circular flows; tachyons; neutrinos; dark matter; black holes

## 1. Introduction

In a recent series of papers I have presented the basis for a theory of tachyons ("faster-than-light" particles) with the idea that neutrinos may be such things. In particular, the Cosmic Neutrino Background (CNB), an enormous sea of low energy neutrinos left over from the Big Bang, may provide novel cosmological effects if they are actually tachyons. One surprising result is that this model readily accounts, both qualitatively and quantitatively for the mysterious phenomenon called Dark Energy. For a survey of that earlier work see Reference [1].

The major purpose of this work is to see if we can also account for Dark Matter, due to the strong gravitational fields that can be produced by low energy tachyons. The challenge is to see if we can theoretically justify a large concentration of CNB in the neighborhood of a typical galaxy. Recent work [2] provided an approximate Hamiltonian for starting that quest. Section 2 is a review of that paper.

It turns out that tachyons with spin 1/2 have their designation as "particle" or "anti-particle" according to their helicity. Furthermore, those two types of tachyon neutrinos contribute to the energy-momentum tensor $T^{\mu\nu}$ with opposite signs. Thus, we imagine that the early CNB, at high energies, mixed both types with little gravitational effects; but, as the universe expanded and cooled, the two types of tachyon neutrinos separated into two different clouds. See the discussion after Equation (3) in Section 2. One type, providing a negative pressure in $T^{\mu\nu}$, remains dispersed throughout the universe and its gravitational effect is seen as "Dark Energy". The other, behaving like a familiar attractive gravitational source, is imagined to have condensed around galaxies and there produce the gravitational effects attributed to "Dark Matter".

In this paper, we report on efforts to find how, in theory, tachyons might be captured into localized orbits. This starts with the mathematics of the Schwarzschild metric: for a static spherically symmetric solution of Einstein's General Relativity. At first, it seems that there will be no bound orbits (only scattering states) for a tachyon. In Section 3, we extend this model allowing for anisotropic pressure, which seems likely to arise from the flow of tachyons, to act upon themselves. This turns out to be a successful search: the possibility of

tachyon bound states is established, qualitatively, in Sections 4 and 5. A program of future quantitative calculations is outlined in Section 6; there we also give numerical estimates that provide a fair quantitative fit to experimental data formerly attributed to "Dark Matter". Beyond that, we are led to look at tachyon orbits in the neighborhood of a Black Hole and find some novel behaviors.

## 2. Hamiltonian Formalism

In a recent paper [2] I derived the following Hamiltonian for the gravitational interaction of a group of particles using the linear approximation to Einstein's full theory of General Relativity and assuming the whole system is static in time. The particles, labeled a,b, have position coordinates $\mathbf{x}$ and velocities $\mathbf{v}$; they may be ordinary $\epsilon = +1$ or tachyons $\epsilon = -1$; and there is another factor $\zeta = \pm 1$ that labels the two helicity states of the tachyons. The other symbols are $\omega = \epsilon \zeta$ and $\gamma = 1/\sqrt{|1 - v^2/c^2|}$. [units: velocity of light $c = 1$.]

$$H = -\sum_a \omega_a m_a \gamma_a - \sum_{a,b} \frac{G(\omega_a m_a \gamma_a)(\omega_b m_b \gamma_b)}{|\mathbf{x}_a - \mathbf{x}_b|} Z_{ab},$$

$$Z_{ab} = 2 - 4\mathbf{v}_a \cdot \mathbf{v}_b + (v_a^2 + v_b^2) - (\epsilon_a \gamma_a^2 + \epsilon_b \gamma_b^2 + 1) \times$$

$$\times [(1 - \mathbf{v}_a \cdot \mathbf{v}_b)^2 - \frac{1}{2}(1 - v_a^2)(1 - v_b^2)]. \tag{1}$$

In the case of low energy ordinary particles this becomes the familiar formula for Newtonian gravity,

$$H = \sum_a (m_a + m_a v_a^2/2) - \sum_{a<b} \frac{G m_a m_b}{|\mathbf{x}_a - \mathbf{x}_b|}. \tag{2}$$

The most interesting case for us is if all the particles are low energy tachyons ($v_a \to \infty$, $\gamma_a \to 1/v_a$):

$$H \to -\sum_{a,b} \frac{G \zeta_a m_a v_a \zeta_b m_b v_b (1/2 - cos^2\theta_{ab})}{|\mathbf{x}_a - \mathbf{x}_b|}., \tag{3}$$

where that angle $\theta_{ab}$ is between the two velocity vectors. We read the sign of this expression as telling us what is gravitational attraction (negative) and what is repulsion (positive). A key factor is the average of the angle $\theta_{ab}$ and this depends on the gross geometry of the distribution of tachyons. For a one-dimensional arrangement of the tachyon velocities (this is the model I examined in my first paper) we see repulsion. For a two dimensional (planar) uniform distribution of velocities we get zero interaction ($< cos^2\theta_{ab} >= 1/2$). For a three-dimensional uniform distribution of tachyon velocities we get attraction ($< cos^2\theta_{ab} >= 1/3$) among same type $\zeta$ but repulsion between opposite types.

This leads to the physical idea that the primordial neutrinos, if they are indeed tachyons, would separate into two sets of configurations each composed of the same helicity attracting one another; but opposite helicity clouds would eventually drift apart.

One more geometry is worth noting: a two dimensional uniform distribution of tachyon velocities over the surface of a sphere. This, the above Hamiltonian says, is a self-attracting arrangement ($<1/2 - cos^2\theta_{ab} >= 1/4 \, sin^2\bar{\theta}_{ab}$, where this $\bar{\theta}_{ab}$ is the angle between the two coordinate vectors). See Appendix A for a derivation of this result. This configuration is taken up in the main part of this paper.

NOTE. There is a minor error in the published paper [2]. In Equation (2.8) of that paper the exponent of the determinant should be $-1/2$ instead of $1/2$. There is also a more complicated criticism about the derivation of the big Equation (1) of this paper but the special case (3) remains valid. My philosophy is that this Hamiltonian may be used to suggest configurations that should be modeled in using more rigorous GR analyses—as in the present paper.

### 3. Modified Schwarzschild Metric

We start by reviewing the standard Schwarzschild metric of General Relativity. For a static spherically symmetric system the metric, with coordinates $x^\mu = (t, r, \theta, \phi)$, is written

$$ds^2 = A(r)dt^2 - B(r)dr^2 - r^2 d\theta^2 - r^2 sin^2\theta \, d\phi^2. \tag{4}$$

For the standard case of a perfect fluid, the energy-momentum tensor is,

$$T_{tt} = \rho \, A, \quad T_{rr} = p \, B, \quad T_{\theta\theta} = p \, r^2, \quad T_{\phi\phi} = sin^2\theta \, T_{\theta\theta}, \tag{5}$$

where $\rho$ and $p$ are functions of the coordinate $r$. From the conservation equation, $T^{\mu\nu}_{\;;\mu} = 0$, we get the relation,

$$p' + (p + \rho)\frac{A'}{2A} = 0. \tag{6}$$

The Einstein tensor is then calculated from the Schwarzschild metric: I copy the results from *B*. Schutz' book, page 260 [3]. He uses a slightly different notation; but after translating, here are the results from $G_{\mu\nu} = 8\pi G T_{\mu\nu}$.

$$m(r) = 4\pi \int_0^r s^2 ds \, \rho(s), \quad B(r) = \frac{1}{1 - 2Gm(r)/r}, \tag{7}$$

$$\frac{A'}{A} = 2G \frac{m(r)/r^2 + 4\pi r \, p(r)}{1 - 2Gm(r)/r} \tag{8}$$

Now I want to show why it is hard to find bound states for tachyons; and then suggest how we might change that. Start with the geodesic equations for the motion of any particle in such a metric. From the $\ddot{t}$ equation and the $\ddot{\phi}$ equation we have the solutions (where dot means $d/d\tau$),

$$\dot{t} = E/A(r), \quad \dot{\phi} = L/r^2; \tag{9}$$

and $E$, $L$ are constants of integration. Then we have the general integral,

$$g_{\mu\nu}\dot{\xi}^\mu \dot{\xi}^\nu = \epsilon, \quad A(r)(\dot{t})^2 - B(r)(\dot{r})^2 - r^2(\dot{\theta})^2 - r^2 sin^2\theta(\dot{\phi})^2 = \epsilon. \tag{10}$$

Given the standard Schwarzschild metric, above, the key relation summarizing the Geodesic equations for a tachyon ($\epsilon = -1$) becomes.

$$B(r)\dot{r}^2 = \frac{E^2}{A(r)} + 1 - \frac{L^2}{r^2} \equiv W(r). \tag{11}$$

(For ordinary particles, that +1 is a $-1$ and a very different analysis follows.) At large $r$ the right hand side is a positive constant; but as we move to lesser values of $r$ the last term takes it negative. If $A(r)$ is a modest function, the right hand side appears to stay negative, as shown by the dotted curve in Figure 1. Thus, we conclude that the particle orbit can never be bound; only scattering states are possible. To change this story into one that allows a bound state, we could ask for a behavior of the function $A(r)$ that makes the right hand side of (11) look like the solid curve in Figure 1.

To get that bump in the expression $E^2/A(r)$ we want the function $A(r)$ to have a sharp minimum at some radius $r = r_1$. If we look at Equation (8) and assume that the pressure term $p(r)$ is the main contributor, then we want to have $p(r_1) = 0$ and $p'(r_1) > 0$. How can that come about? In Section 2, studying the Hamiltonian formalism, it was suggested that a circular flow of tachyons could be an attractive idea for modeling.

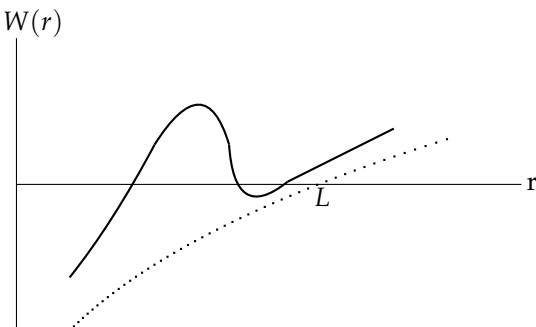

$W(r)$

$L$

r

**Figure 1.** The function *W(r)* for a tachyon: two models.

Therefore, I now want to introduce a modified Schwarzschild model. It is assumed above that the pressure, i.e., the flow of stuff that gives rise to the energy-momentum tensor, is isotropic: that is, the flow is the same in all directions at each point in space. That is reasonable for a gas of ordinary matter, where there are many rapid, and random, scattering events among the particles. That is sensible for models of stars. However, my interest is in a sea of neutrinos, specially described as low energy tachyons. These particles hardly interact at all with anything else at short distances; their main interaction is with the gravitational field at large distances. Thus, one would imagine that the flow is a complicated function of space: definitely not isotropic. To make the mathematics simpler I shall stay with the assumption of spherical symmetry; and this leads me to say that there are two components of the "pressure": $p_r$ for the flow in radial directions and $p_a$ for flow in angular directions. These are each functions of the coordinate r. The input for this modified problem is thus different from the equations above:

$$T_{tt} = \rho\, A, \quad T_{rr} = p_r\, B, \quad T_{\theta\theta} = p_a\, r^2, \quad T_{\phi\phi} = sin^2\theta\; T_{\theta\theta}. \tag{12}$$

The solutions for the metric functions *A(r)* and *B(r)* are only slightly changed:

$$m(r) = 4\pi \int_0^r s^2 ds\, \rho(s), \quad B(r) = \frac{1}{1 - 2Gm(r)/r}, \tag{13}$$

$$\frac{A'}{A} = 2G\, \frac{m(r)/r^2 + 4\pi r\, p_r(r)}{1 - 2Gm(r)/r}; \tag{14}$$

and the main change appears in the equation from the conservation of *T*:

$$p_r' + (p_r + \rho)\frac{A'}{2A} = -\frac{2}{r}(p_r - p_a). \tag{15}$$

I need not present a detailed derivation of this equation because there is a substantial literature using this formalism of anisotropic pressure in the Schwarzschild metric to model the structure of stars [4]. We are interested in tachyons, something rather different.

If one sets $p_r = p_a$, then this reduces to the earlier equation; and such equality is the statement of isotropic flow. It will be best to rewrite those variables as: $p_r(r) = p(r)$ and $p_a(r) = p(r) + \sigma(r)$. Thus, $p(r)$ is the isotropic flow at each radius and $\sigma(r)$ is the *additional* angular flow at each r. The basic equation relating them is now,

$$p' + (p + \rho)\frac{A'}{2A} = \frac{2\sigma}{r}. \tag{16}$$

Let us look at the simplest model: where we set $\rho = 0$ and study only the effect of the two pressure components.

### 4. Pressure Only

The equations we have are,

$$B = 1, \quad \frac{A'}{A} = 2\,b\,r\,p(r), \quad b = 4\pi G, \tag{17}$$

$$p' + b\,r\,p^2 = \frac{2\sigma}{r}. \tag{18}$$

This last equation says that we cannot have any extra angular pressure $\sigma$ without some isotropic pressure $p$. I take this as a reminder that the conservation of energy theorem $T^{\mu\nu}{}_{:\mu} = 0$ is not just about particles and other sources, it is also about the gravitational field itself. We should not try too hard to separate those components in our model building.

If we imagine that $\sigma(r) = 0$ in some region of r, then the solution there is $p_0(r) = 2/[br^2 + constant]$; or, alternatively, $p(r) = 0$. The simplest model is to insert one or more delta functions for $\sigma$, the added angular pressure. That would imply adding thin shells of particles each moving in circular orbits uniformly distributed over a sphere of some given radius. (This picture was suggested in Section 2.)

$$\sigma(r) = \Sigma_1\,\delta(r - r_1) - \Sigma_2\,\delta(r - r_2). \tag{19}$$

The solution of the differential equation is then,

$$p(r < r_1) = 2/[b(r^2 - r_1^2) - c], \quad p(r_1 < r < r_2) = 2/[b(r^2 - r_1^2) + d],$$
$$p(r > r_2) = 0, \quad [1/d + 1/c] = \Sigma_1/r_1, \quad 1/[b(r_2^2 - r_1^2) + d] = \Sigma_2/r_2. \tag{20}$$

The constants c and d should be greater than zero, to avoid any singularities. I have forced the pressure to be zero outside of $r = r_2$ as a boundary condition. Figure 2 shows what a solution of this sort looks like when c is slightly smaller than d. (We here assume $\Sigma_1 > 0$; the alternative may be considered.)

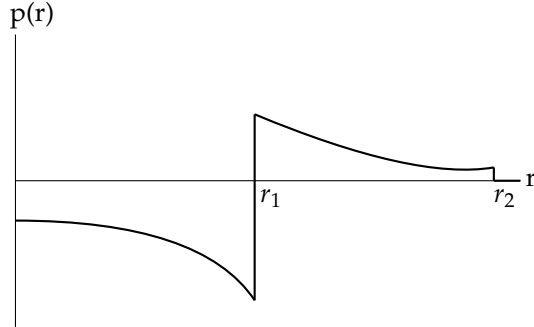

**Figure 2.** Solution *p(r)* from Equation (20).

It is tempting to read this picture in familiar physical terms: We put in a circular flow of matter around the surface of a certain sphere and got out some radial pressure that is negative inside that sphere (i.e., "pulling" the rotating particles toward the center) and positive outside that sphere (i.e., "pushing" the rotating particles toward the center). However, I should caution myself that General Relativity is frequently much more difficult to understand; such familiar interpretations may be misleading.

Alternatively, we might read these equations the other way: We put an isotropic pressure into some volume of space and we get out, from Equation (18), the requirement that there be some additional angular pressure, even if the isotropic pressure is uniformly distributed. (i.e., $p' = 0$ does not imply $\sigma = 0$.) This is another example of what I would call the "attribution problem"—trying to assign familiar physical meaning to terms that we find in the equations of General Relativity.

In other well known solvable problems in General Relativity it appears that the gravitational field itself must contribute to the energy-momentum tensor, in addition to the presence and motion of massive particles (and radiation). The textbooks tell us that the standard Schwarzschild solution leads to the definition $M = 4\pi \int_0^\infty r^2 dr\, \rho(r)$, where $\rho$ is defined as something given in $T_{tt}$; but we should not interpret this M as being the total mass of the source. The Robertson-Walker model for the universe starts with some quantities $\rho, p$ written down in $T^{\mu\nu}$ and we casually assume that these represent actual particles and fields; but is that really true? The most extravagant example of the "attribution problem" is, of course, the cosmological constant. The dominant theory predicts a value that exceeds the observations by a factor of $10^{120}$. A classic review of this topic may be found in the paper by Weinberg [5]. I do not fully understand all of this but I shall just proceed as best I can. My mission here is one of modeling. I take the functions $p(r), \sigma(r)$, and even $\rho(r)$, as the clay I am free to play with and shape as I wish—subject only to the mathematical requirements of Einstein's theory of General Relativity. Physical interpretations can wait.

Next we can calculate the function A(r) by using the Formula (20) to integrate Equation (17) from the limits $r_2$, *to* $r$. This will assure the boundary condition $A(r > r_2) = 1$. Figure 3 shows what the result looks like. This is the sort of shape we sought to get tachyon neutrino bound states.

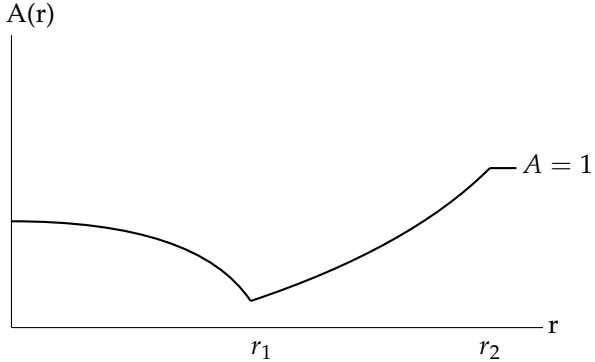

**Figure 3.** Solution *A(r)* with the input *p(r)* as in Figure 2.

The formulas for the graph in Figure 3 are as follows, with the dimensionless parameters, $\alpha_1 = br_1^2/c$, $\alpha_2 = br_1^2/d$.

$$A(r < r_1) = A(r_1)[1 + \alpha_1(1 - r^2/r_1^2)]^2, \tag{21}$$

$$A(r_1 < r < r_2) = A(r_1)[1 + \alpha_2(r^2/r_1^2 - 1)]^2, \tag{22}$$

$$A(r_1) = [1 + \alpha_2(r_2^2/r_1^2 - 1)]^{-2}, \quad A(r > r_2) = 1. \tag{23}$$

We see that $A(r_1)$ might be very small for large $r_2/r_1$ even if $\alpha_2$ is rather small.

We would like to go on to examine solutions that have the circulating flow spread out somewhat in a radial distribution; but with a nonlinear differential equation we must do more than just add solutions of this simple type. An easy way is to guess a likely shape of the function p(r), then calculate A(r) from Equation (17) and use Equation (18) to solve for $\sigma(r)$. One example is $p(r) = a(r_1^2 - r^2)(r^2 - r_2^2)$, with $p(r > r_2) = 0$; this leads to $A(r_1) = exp[-ab(r_2^2 - r_1^2)^3/6]$, which can be very small compared to $A(r \geq r_2) = 1$.

### 5. Add Mass to Pressure

Now we enlarge this analysis by including the terms involving $\rho$ along with our two components of the pressure.

$$B(r) = 1/A_0(r), \quad A_0(r) \equiv 1 - 2Gm(r)/r, \quad m(r) = 4\pi \int_0^r s^2 ds \rho(s), \tag{24}$$

$$\frac{A'(r)}{A(r)} = \frac{A_0'}{A_0} + \frac{2br(p_r + \rho)}{A_0}, \quad p_r' + \frac{A'}{2A}(p_r + \rho) = \frac{2}{r}\sigma, \quad b = 4\pi G. \tag{25}$$

Following our earlier path, we set $p_a = p_r + \sigma$ and $p_r(r) + \rho = q(r)A_0^{-1/2}$. This leads to

$$q'(r) + \alpha(r)q(r)^2 - \rho' A_0^{1/2} = \beta(r)\sigma(r), \quad \alpha = brA_0^{-3/2}, \quad \beta = (2/r)A_0^{1/2}. \tag{26}$$

In any region of r where $\sigma = \rho = 0$, we have the solution,

$$q_0(r) = \frac{1}{\eta(r) + constant}, \quad \frac{d\eta}{dr} = \alpha(r); \quad or \quad q_0(r) = 0. \tag{27}$$

This all looks very similar to what we saw in the previous section. If we look at large $r$, we see $\eta \sim r^2$ and thus we have to pay attention to boundary conditions at large $r$.

Here is a simplified model that I will use for studying the gravitational structure of galaxies, with a large presence of tachyons. I assume that $\rho$, the energy density term in the energy-momentum tensor, is small compared to the pressure terms; but I will keep the function $A_0(r)$ to represent all of that.

$$A(r) = A_0(r)A_1(r). \tag{28}$$

Then, I assume that the pressure terms are dominant in determining the rest of the information about the metric: the function $A_1(r)$. The results, further simplified by setting $A_0 = 1$ in these equations, are:

$$\frac{A_1'}{A_1} = 2brp_r, \quad p_r' + brp_r^2 = \frac{2}{r}\sigma, \tag{29}$$

which is just the equation we studied in Section 4. So, I will take Figure 3 as a typical solution for $A_1(r)$, remembering that the full metric function is $A = A_0 A_1$.

### 6. Is Dark Matter Really Due to Tachyon Neutrinos?

Dark Matter is the phrase used to describe a mysterious source of gravitational fields observed in typical galaxies. The relevant experimental data are generally of two types: excess velocity of stars as a function of distance from galactic center and gravitational lensing. The most popular theories for explaining those phenomena are based upon Newtonian theory of gravitation and posit some massive, but unseen, particles dispersed throughout the universe. Extensive searches for such things have turned out to be fruitless.

What I have been considering is the possibility for the large collection of relic neutrinos (CNB) to be identified as tachyons, with a mass perhaps around 0.1 eV. The earlier studies of this idea showed that, using Einstein's Theory of General Relativity, this source could generate unexpectedly large gravitational fields, especially at very low energies, through their particular contribution to the spatial components of the energy-momentum tensor. In one simple calculation, it was found that this could explain—both qualitatively and quantitatively—the other mysterious phenomenon called "Dark Energy".

As mentioned in Section 2 of this paper, there would be two types of tachyon neutrinos (differing in their helicity) that contribute to the energy-momentum tensor with opposite signs.Thus, an attractive idea was that one type could remain dispersed throughout the universe (giving Dark Energy) while the other type could condense upon galaxies and give us Dark Matter.

As laid out in Section 3, it is at first hard to see how tachyons might be localized by gravity if we stick with simple Schwarzschild models. However, the consideration of large anisotropic flows of matter (especially such as we expect from low energy tachyons) led us to find that, indeed, bound orbits were feasible. Thus, we claim to have found a qualitative model for explaining Dark Matter. What remains is to show that this theory is quantitatively viable.

My first instinct was to say that we need a program of future calculational work following the ideas presented in this paper. I could describe it as something like the program of self-consistent fields, the Hartree–Fock model, for atoms. Construct a plausible model of the pressure distribution (as illustrated above), this leads to a specific form for the metric A(r); fill the available bound states with tachyon neutrinos; calculate the energy-momentum tensor from that; proceed to adjust the input pressure distribution; repeat. However, I remember the discussion following Figure 2 about the "attribution problem". We cannot totally control what should go into the energy-momentum tensor in terms of our image of particles as distinct from the gravitational field itself. So I will proceed as best I can to find some quantitative measures within this modeling.

I can give a numerical estimate of the critical parameter involved in these models. In the analysis at the end of Section 4 we saw a dimensionless parameter, which I will call $\lambda_1$ that indicates the size of the "dip" in the metric function $A(r)$ or $A_1(r)$ (see Figure 3), which translates into the size of the "bump" in the term $E^2/A(r)$ (see Figure 1 or Figure 5) that is critical in the geodesic equation allowing bound states for tachyons.

$$\lambda_1 = 4\pi G \, \bar{p} \, R^2/c^4. \tag{30}$$

Here $\bar{p}$ is the magnitude of the pressure term put in the energy-momentum tensor and R is the size of the tachyon cloud attached to the galaxy. In earlier work I used a value for the pressure of tachyon neutrinos in the CNB, call that $p_{CNB} \sim 10^5$ eV/cm$^3$. (This substitutes for the Cosmological Constant that is used in other theories.) If my model says that a great portion of the CNB is condensed into something attached to the galaxies, then I would estimate $\bar{p} = p_{CNB}(D/R)^3$, where D is the typical distance between galaxies. If I set R equal to the typical size of a galaxy, then I get a value for $\lambda_1$ that is of the order of $10^{-6}$.

Note. At first I worried that condensing the CNB (that factor of $(D/R)^3$) would lead to an increase in the tachyons' energy, thus reducing their velocity and also their pressure. However, referring to an earlier paper [6] where I studied statistical mechanics for tachyons, I found that, for low energy tachyons, the pressure remains proportional to the number density.

That $\lambda_1$ looks like a rather small number. There is substantial uncertainty here, but we must ask whether such a small parameter value negates the objective of finding a full explanation for Dark Matter in the tachyon neutrino theory. The primary experimental observations that led to the postulate of Dark Matter was the study of the excess velocity of stars moving within a galaxy, especially when plotted as a function of coordinate r from the center. Ordinarily, Newtonian theory of gravitation should explain the orbits of such stars; and the visible matter throughout the galaxy was used as indicating the source of such gravity. Let me write another dimensionless parameter, call it $\lambda_0$ to measure this Newtonian gravity.

$$\lambda_0 = \frac{G \, M}{c^2 \, R}, \tag{31}$$

where $M$ is the total mass of the ordinary matter in the galaxy. Putting numbers into this I get $\lambda_0$ is of the order of $10^{-6}$. [Hmmm.]

We can now write a representation of the metric function A(r) as follows.

$$A(r) = A_0(r)A_1(r) = [1 - \lambda_0(R/r)][1 - \lambda_1 f(r/R)] \approx [1 - \lambda_0(R/r) - \lambda_1 f(r/R)] \tag{32}$$

The function *f(r/R)* should be a positive function that decreases toward zero as its argument grows toward 1. That is the description of the results from the previous section.

The motion of non-relativistic particles in Einstein's GR reduces to Newtonian motion in Newtonian gravity with the identification $A(r) = 1 + 2V(r)$ and V is the familiar Newtonian potential. The equation above shows us two components of the total potential: one from ordinary matter and the other from our model of tachyon neutrinos. *They are both of the same order of magnitude; and they have different behaviors as functions of r!* The circular motion of stars in the galaxy is described by the elementary physics formula $-F/m = V' = v^2/r$. The experimental data is nicely displayed in the graph found in the Wikipedia article on Galaxy Rotation Curve. The dashed curve shows what is expected from Newtonian gravity due to the visible distribution of ordinary matter in the galaxy. That is from $A_0$. The solid curve shows the observed behavior of velocity as a function of r. The distance between those two curves is commonly said to be due to "Dark Matter". Our theory of tachyon neutrinos gathered around the galaxy gives an account of this difference, through $A_1$, that is qualitatively and quantitatively fitting.

Here is the simplest model: $f(r/R) = 1 - r^2/R^2$. This gives $-V' = \lambda_1 r/R^2$ and thus the galaxy velocity curve

$$v^2 = \lambda_0\, R/r + \lambda_1\, r^2/R^2, \tag{33}$$

which looks like a very good fit to the experimental data. Furthermore, from Equation $A_1'/A_1 = 2brp$, this model says that $p(r) = constant$, which is the simple picture of the sea of CNB tachyon neutrinos condensed, pretty much uniformly, about the galaxy. (However, again facing up to the attribution problem, it is not nice to interpret the angular pressure given by this simple model.)

The second set of observations commonly ascribed to Dark Matter is gravitational lensing—the bending of light rays from distant sources as they are deflected by gravitational fields in galaxies. Within the standard theory of General Relativity (and I stay loyal to this, while some other theorists try to change it) all this must be due to the one and only metric in space-time. If we have now successfully explained the galaxy rotation curve with the theory of tachyon neutrinos—and explicitly constructed the modified Schwarzschild metric to do so—then the explanation for gravitational lensing is now also achieved.

Two expert reviewers of this manuscript have asked that I elucidate the remarks of the last paragraph. The currently prevailing theory of Dark Matter is denoted by the acronym CDM (Cold Dark Matter), although the earlier acronym WIMP (Weakly Interaction Massive Particles) is more descriptive. It allows Newtonian theory of gravitation to be used, which is a lot easier than Einsteinian gravitation. The idea is that some mysterious type of heavy particle, which does not show itself to astronomers, is sprinkled about the galaxies in some distribution that adds to the gravitational field expected from the visible matter in the galaxy and thus is responsible for both the Galactic Rotation data and the Gravitational Lensing data. I assume that there are detailed models constructed in that theory that show that both of those goals can be consistently achieved. I assert, in the preceding paragraph, that having shown that my alternative theory of tachyon neutrinos can explain the one set of data, then the other set of data is also explained. The basis for this claim is that what the rotating stars in the galaxy respond to and what the photons being bent in their path by the galaxy respond to is not the Cold Dark Matter but rather the one and only total gravitational field—represented in Einstein's General Relativity by the space-time metric. My theory gets the metric to do the one job; so I have the correct metric; and it does both jobs. There is further data on galactic clusters that is also ascribed to Dark Matter; I have said nothing about that topic, which requires more complicated calculations than my work within the Schwarzschild model of spherical symmetry allows.

There is one other area in the model presented above that needs to be looked at: what if we get close to a black hole, where the approximation of weak behavior for $A_0(r)$ is unsupportable.

### 7. Tachyons near a Black Hole

We want to look at the geodesic equation for tachyons: $\epsilon = -1$ in the neighborhood of a black hole. At first we ignore any contributions of pressure and just use the Schwarzschild metric.

$$(\dot{r})^2 = U(r), \quad U(r) = \frac{1}{B(r)}\left[\frac{E^2}{A(r)} - \frac{L^2}{r^2} + 1\right], \tag{34}$$

$$A = A_0 = (1 - r_s/r), \quad B = 1/A_0, \quad r_s = 2GM, \tag{35}$$

$$U(r) = E^2 + (1 - r_s/r)(+1 - L^2/r^2); \tag{36}$$

Let us take a closer look at this effective potential $U(r)$. The two factors on the right give us zeros at $r = r_s$ and at $r = L$; and this product is negative in between those two points. See Figure 4, where we assume that $L > r_s$ and that the constant $E$ is either zero (the dotted line) or small (the solid line).

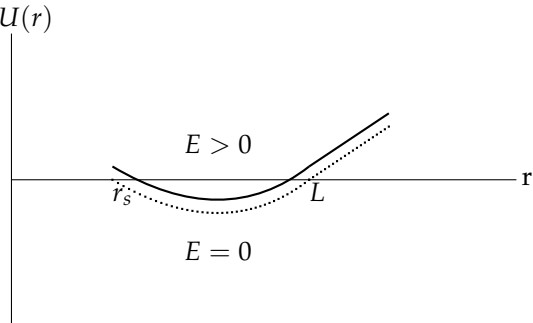

**Figure 4.** The effective potential *U(r)* for a tachyon outside of a black hole.

The region where the graph of $U(r)$ is negative is forbidden for the particle trajectory. So, in Figure 4, we see scattering states that start at infinity and reach a closest approach at $r = L$ (dotted line), or slightly closer (solid line). For ordinary particles near, but outside of a black hole, there is the possibility of bound orbits. For tachyons, it was shown some time ago, that no such stable orbits exist outside of the black hole [7].

Now we add in the effects of the pressure, as studied in the previous section. The term $E^2$ in Equation (36) is now replaced by $E^2/A_1(r)$, where the function $A_1(r)$ is, for example, displayed in Figure 3. If we choose the parameters in that model appropriately, we may get the modified graph of U(r) shown in Figure 5. With $A_1$ diving to very small value in the middle, $E^2/A_1$ shows an upward bump. This is now a picture that tells us, YES, there can be stable localized stable orbits for tachyons within this bump. Thus, we are able to combine a black hole and a particular arrangement of circular flows to shape the necessary metric for tachyon bound states.

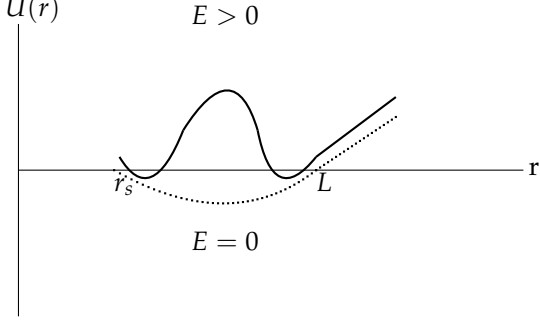

**Figure 5.** The effective potential *U(r)* for a tachyon outside of a black hole, with pressure.

As an alternative to the picture in Figure 4 we can consider the case $r_s > L$ and get the picture in Figure 6 (without effects of the pressure). This leads us to immediately ask about how tachyons travel inside a black hole, when they come in with a small impact parameter. If the value of E is large, as shown in Figure 6, we may imagine that the tachyon comes in from infinity, slows down somewhat inside the black hole, but then emerges from another part of the black hole and escapes out to infinity again. That is a higher energy scattering phenomenon—provided that what we just said about tachyon trajectories inside a black hole is correct. On the other hand, if the value of E is small enough so that the upper curve in Figure 6 dips below $U = 0$, then we might imagine the possibility of a localized bound state for the tachyon orbiting entirely inside the black hole.

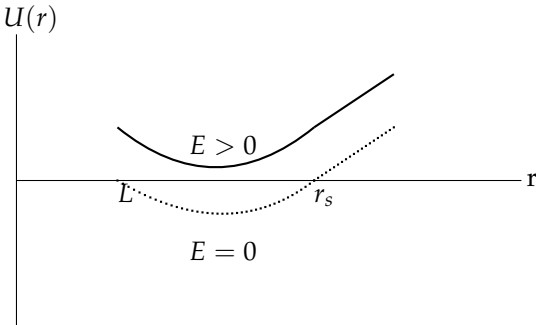

**Figure 6.** The effective potential *U(r)* for a tachyon impacting a black hole.

The standard knowledge about black holes is that any ordinary particle, or light, traveling into a black hole, passing the horizon at $r = r_s$, will be sucked into the physical singularity at $r = 0$ and never come back. Is that true as well for tachyons? Has anyone investigated that question before?

To look inside the horizon of a black hole $r < r_s$, we need to work with something other than the exterior Schwarzschild solution. In the next section we shall investigate the inner geometry of a black hole using the Kruskal-Szekeres (K-S) coordinates.

## 8. Particle Trajectories Inside a Black Hole

Particle trajectories are determined by the geodesic equations, given a metric $g_{\mu\nu}(x)$, and this is an integral derived from those equations in general.

$$g_{\mu\nu}\dot{\zeta}^\mu\dot{\zeta}^\nu = \epsilon, \tag{37}$$

where $\epsilon = +1$ for ordinary particles ($v < c$), $\epsilon = 0$ for light ($v = c$), and $\epsilon = -1$ for tachyons ($v > c$). If we are given the Schwarzschild metric for a static, spherically symmetric system, this reads,

$$A(r)(\dot{t})^2 - B(r)(\dot{r})^2 - r^2(\dot{\Omega})^2 = \epsilon, \quad A = B^{-1} = (1 - 2GM/r), \tag{38}$$

where we sit outside of the source of mass *M*; and the "horizon" is at the Schwarzschild radius $r_S = 2GM$.

Experts tell us that to study the solutions inside the event horizon, $r < r_s$, we should use the Kruskal-Szekeres (K-S) coordinates. I shall not repeat everything about that subject found in textbooks [3], but just one relevant formula and a picture. The coordinates (r,t) are mapped into new coordinates (u,v) while the angular coordinates $(\theta, \phi)$ remain as before. For the angular motion we still have the solution $\dot{\phi} = L/r^2$.

The picture, Figure 7, is a standard representation of the K-S variables. The coordinate singularity at the horizon $r = r_s$ is mapped into the 45-degree lines $u^2 = v^2$ and the physical singularity at $r = 0$ is now mapped to the heavy hyperboloid in the top quadrant, labeled region II, which is the inside of the black hole. The quadrant to the right ($|v| < u$), labeled

region I is the familiar region outside of the event horizon. The key geodesic equation becomes,

$$e^{-r/2r_s} \frac{4r_s^3}{r} (\dot{v}^2 - \dot{u}^2) - \frac{L^2}{r^2} = \epsilon = (+1, 0, -1). \tag{39}$$

At every point in the (u,v) plane the light-cone looks as it does in flat space. Thus, ordinary particles are expected to move mostly in the vertical direction while low energy tachyons would be expected to move mostly in horizontal directions.

If we look at the geodesic Equation (39) for ordinary particles $\epsilon = +1$ or for light $\epsilon = 0$, we see $\dot{v}^2 - \dot{u}^2 \geq 0$. This means that any such particle, initially outside the event horizon but moving across that line, will inevitably crash into the singularity at $r = 0$. This is how black holes eat up ordinary matter and radiation. Figure 7 shows this: the dotted line is the trajectory of an ordinary particle, coming in from outside, crossing the event horizon, and then diving into the center $r = 0$.

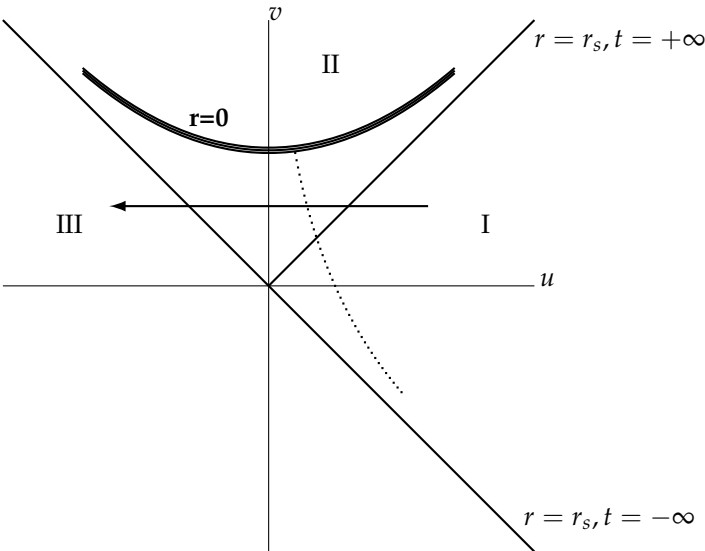

**Figure 7.** K-S coordinates for a black hole. The dotted line is the trajectory of an ordinary particle that crosses the event horizon and is consumed at the center. The arrow line shows a possible transit of a tachyon through the black hole.

Now we look at the possible trajectories of a tachyon inside a black hole. We have a different looking constraint:

$$e^{-r/2r_s} \frac{4r_s^3}{r} (\dot{v}^2 - \dot{u}^2) = \frac{L^2}{r^2} - 1. \tag{40}$$

The expression on the right may be positive or negative. For $r > L$ the right hand side is negative, and so we expect the trajectory to be dominated by motion in the horizontal direction, parallel to the u-axis. If the tachyon comes from outside (region I) and crosses the horizon $r = r_s$ into region II, we expect it to travel on to the left and exit the black hole by crossing that other 45 degree line into that other outside, region III. This is a scattering process, as indicated in Figure 6 for large values of $E$.

Might tachyons find stable localized orbits inside a black hole? The discussion around Figure 6, combined with the analysis of this section, suggests that it might. However, when we do more careful calculations in Appendix B, we find that this is not so.

## 9. Summary of Results and Further Questions

The purpose of this study was to see if we could find any possibility of stable localized orbits for tachyons within a galaxy. This was successful in two ways.

The first model, with orbits reaching out perhaps to cover a large fraction of the galaxy, required the assumption of large circulating flows of tachyons to create the pressure configurations that kept those orbits in place. Figures 1 and 3 are a representation of that scheme.

The second model looked in the close neighborhood of a black hole and found similar possible bound orbits (see Figure 5). This model has orbits that may penetrate within the black hole. We relied upon the transition from Schwarzschild coordinates to the Kruskal-Szekeres coordinates to see how tachyons, inside a black hole, behave very differently from ordinary particles. Here we ignored the effects of pressure in shaping the metric.

In Section 6, we put some numbers into the qualitative formulas derived and showed that the model of tachyonic neutrinos condensed into bound orbits about a typical galaxy would produce the effects on stellar orbits that were previously attributed to some mysterious Dark Matter. Further calculations are needed to make this fitting to the data more precise; but it appears that we have achieved a major discovery with this theory.

There is one more numerical estimate that I should make here and that has to do with the Fermi statistics for neutrinos. In the standard model of the CNB the density of each flavor of neutrino is found to be 56 cm$^{-3}$. That means that the average distance between any two is about $d = 0.26$ cm. To estimate the effect of the Pauli exclusion principle, we may compare this distance to the Compton wavelength $h/mc = 1.2 \times 10^{-3}$ cm using a neutrino mass of 0.1 eV. The ratio of these two numbers is about 200. In Section 6, I condensed the CNB by a linear factor $D/R$ of 10; that reduces this ratio to about 20. I conclude that this condensation is not impeded by the Fermi statistics. (However, it is getting close. So this is a subject for future study.)

There is plenty of good work that needs to be undertaken, following the results presented in this paper, to refine the simple calculations found above and to see other implications in cosmology for the neutrino-tachyon theory. The most obvious next study should revise the FLRW model of the evolution of the universe using tachyon neutrinos, as described in this paper, instead of Dark Energy and Dark Matter in the $\Lambda CDM$ model. If I had any graduate students working under me I would already have assigned them to this work. However, I am retired for three decades and have no such resources. So I invite any reader to take up this study.

Further questions are: How do the tachyon neutrinos get from the CNB into these orbits inside a galaxy? Going beyond what I have covered here, someone should look into the Kerr metric, used to describe a rotating black hole, and see how tachyons behave there.

**Funding:** This research received no external funding.

**Data Availability Statement:** Not applicable.

**Conflicts of Interest:** The author declare no conflict of interest.

## Appendix A. Average over Velocities on a Sphere

We want to evaluate $<1/2 - cos^2\theta_{ab}>$ where the average is taken over a uniform distribution of velocities on the surface of a sphere and $\theta_{ab}$ is the angle between any pair of velocities.

Introduce the unit vectors in spherical polar coordinates as they relate to Cartesian unit vectors.

$$\hat{\theta} = cos\theta \; (cos\phi \; \hat{x} + sin\phi \; \hat{y}) - sin\theta \; \hat{z}, \quad \hat{\phi} = cos\phi \; \hat{y} - sin\phi \; \hat{x}. \tag{A1}$$

Now we write one velocity vector located at the north pole and oriented in the x-direction: $\mathbf{v}_a = v \; \hat{x}$; and the other at coordinates $(\theta, \phi)$ oriented at an angle $\alpha$ in the tangent plane: $\mathbf{v}_b = v(cos\alpha \; \hat{\theta} + sin\alpha \; \hat{\phi})$. Now we take the dot product.

$$\mathbf{v}_a \cdot \mathbf{v}_b = v^2[cos\alpha \; cos\theta \; cos\phi - sin\alpha \; sin\phi]. \tag{A2}$$

Now, square this expression and average over the angle $\alpha$; then average over the angle $\phi$.

$$< cos^2\theta_{ab} >_{\alpha,\phi} = \frac{1}{4}(cos^2\theta + 1). \tag{A3}$$

This yields the stated result.

**Appendix B. Details on Particle Motion Inside a Black Hole**

Here are some more details on the subject of Section 8.

In the Schwarzschild coordinates, valid outside of a BH, the geodesic equations give one further result, not mentioned earlier:

$$\dot{t} = E/A(r), \quad A(r) = (1 - r_s/r), \tag{A4}$$

where $E$ is a constant of integration. If we start with a particle (ordinary or tachyon) approaching the horizon $r = r_s$ from outside, then we would choose $E > 0$.

What is the analogy of this equation valid inside the BH? I assert that this same Equation (A4) is correct but we now need to understand the coordinates r and t as they are represented in terms of the Kruskal-Szekeres coordinates. (What remain the same inside are the constant $E$ and the affine parameter $\tau$, where the dot means $d/d\tau$.) Therefore, once the particle crosses into the BH, Equation (A4) says that the time coordinate t must now be decreasing!

Let us look at this in terms of the K-S diagram in Figure 7. Again, from textbooks (or from the Wikipedia article about the K-S coordinates) it is shown where the lines of constant r or constant t are in this diagram. Lines of constant r are hyperbolas in each region: in region I, $r$ gets smaller as one moves to the left; in region II, r gets smaller as one moves upwards. (The boundary between those two regions is $r = r_s$,) The lines of constant $t$ are radial straight lines from the origin $u = v = 0$ in each region. In region I t goes from $-\infty$ at the lower boundary to $+\infty$ at the upper boundary. In region II t goes from $+\infty$ at the right boundary to $-\infty$ at the left boundary.

Let us put this all together. An ordinary particle enters the BH traveling somewhat vertically from region I to region II. It reaches the horizon $r = r_s$ at $t = +\infty$. This is a familiar result and textbooks remind us that this time coordinate t is the one used by an observer far away from the BH; so one should not get upset about this infinity. Once inside, before it hits the singularity at $r = 0$, it's trajectory will cross lines of constant t in a sequence that will be read as decreasing t. So we might say that it reaches $r = 0$ (and dies) at some finite time t, that is earlier than when it crossed into the BH. Oh, well, this is a reminder that one should not try too hard to give "physical" meaning to this variable t inside a BH.

Now, a tachyon entering the BH follows a different path in the K-S diagram. It moves from region I to region II in a somewhat horizontal trajectory. It must still pass $t = +\infty$ but then it can avoid the singularity at $r = 0$ and pass out into region III. During this passage, it will cut lines of constant t uniformly decreasing from $t = +\infty$ to $t = -\infty$. When its trajectory returns to the outside of the BH, the time variable is at $t = -\infty$ and then increases. That appears to be entirely consistent with Equation (A4). That whole process is described as a scattering of a tachyon by a black hole, with the particle penetrating inside the horizon and then exiting. (However, as to interpreting this story in terms of the variable $t$: "fuggetaboutit".)

To study the purported bound state for a tachyon inside a BH we need to do some more work. Let us write down the new coordinates inside the BH ($r < r_s$).

$$u = w(r)sinh(t/2r_s), \quad vs. = w(r)cosh(t/2r_s), \quad w(r) = (1 - r/r_s)^{1/2}e^{r/2r_s}. \tag{A5}$$

Now take the derivative of these new coordinates with respect to the affine parameter $\tau$, where r and t are functions of $\tau$; and calculate,

$$\dot{v}^2 - \dot{u}^2 = -w^2\dot{t}^2/4r_s^2 + (w')^2\dot{r}^2. \tag{A6}$$

We put this into Equation (39) and get

$$A\dot{t}^2 - \dot{r}^2/A = \epsilon + L^2/r^2. \tag{A7}$$

where $A = (1 - r_s/r)$ from earlier. This reads just like the original equation in the Schwarzschild coordinates (10); except this equation is valid inside the BH, where $A$ is negative.

For ordinary particles ($\epsilon = +1$) this has no surprises; the $\dot{r}$ term must dominate the $\dot{t}$ term in order to keep the right hand side positive. So the particle trajectory will proceed to $r = 0$.

For tachyons ($\epsilon = -1$) the right hand side will be negative for $r > L$. We already have $r < r_s$ (we are inside the BH, in region II of the picture Figure 7), so this equation tells us that the $\dot{t}$ term must dominate. That means that the particle moves relentlessly to the left in region II while staying away from $r = 0$. This is the penetrating scattering process described earlier, but here with more rigor. (see the arrow line in Figure 7).

For $r < L$ the right hand side is positive. Thus, the $\dot{r}$ term must dominate and we conclude that any such orbit will spiral in to $r = 0$. There is no tachyon (stable) bound state inside a Black Hole. (What if we want to consider some pressure effects inside the BH? Then a new K-S solution may be needed.)

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
