# Peer review of "Theoretical Search for Gravitational Bound States of Tachyons"

_2571-712X, doi:10.3390/particles5030027_

Round 1

Reviewer 1 Report

This is an interesting and innovative investigation into the behavior of tachyons in a gravitational field, with the main objective of seeing whether, if neutrinos are tachyons, the CNB can account for the properties of dark matter. A definitive answer is not given, but some of the results are encouraging. I believe this paper is worthy of publication, after a few small items are dealt with:

1)    In section 2 (line 61 on page 2) the author considers a situation in which the velocity vectors are confined to the surface of a sphere, and produces a result in which the average of the cosine squared of the angle between velocity vectors is related to the average of the sine squared of the angle between coordinate vectors. This is sufficiently mysterious that the author should provide either a derivation or a reference;

2)    On page 9, lines 216-222, the author claims that, having solved the galactic rotation problem, the gravitational lensing problem automatically follows. I don’t understand the logic. Is it not possible that the distribution of tachyonic matter necessary to explain galactic rotation curves is different from the distribution of ordinary dark matter that would be required, and that the latter could give the observed lensing whereas the former does not?

3)    The author has concentrated on one property of dark matter – its influence on galactic rotation (and possibly lensing, modulo comment 2). But dark matter is also invoked to explain the binding in galactic clusters, which is a different problem, and the evolution of structure in the universe, which requires the dark matter to be “cold” at the time of structure formation. Perhaps it is premature, given the state of the current investigation, to try to deal explicitly with these issues, but they should be acknowledged.

Reviewer 2 Report

The author of the paper under review presents a model in which dark matter can be understood by the presence of strong gravitational fields produced by low energy tachyons. In particular the author studies the case in which a large collection of relic neutrinos behave as tachyons. These assumptions, according to the author, provide some evidence for the tachyons to account for dark matter in a scenario where there is a non isotropic  pressure, showing the possibility that such particles would produce similar effects on stellar orbits than those attributed to Dark Matter. Also tachyonic particles are considered around a Schwarzschild Black Hole for which the author reports that possible bound orbits are found.

Although the manuscript is well written, clear and the conclusions are well related to the assumptions and initial objectives, I find some reasons to not accept the paper at this form. 

1. Dark Matter hypothesis is very robust. Reproducing stellar orbits is just one, among many other features, to be considered by any alternative model. Although it is also mentioned that Dark matter implies the presence of lensing effects, it is not clear how tachyonic matter could also reproduce such an effect. I would suggest to add some detailed comments about the plethora of secondary effects that Dark Matter can in principle explain. Very important is to reproduce that 27% of total energy and matter according to the current cosmological standard model. I would like to read some comments about it (not a solution, just comments)

2. Although the author has a lot of experience studying tachyonic matter, there are many other references that I would like to be cited in a manuscript submitted to a special issue. The number of references is quite poor implying that an expected review of the state of art is also very limited.

3. I´m confused about the main result of a tachyonic particle crossing the the Horizon of a Black Hole in K-S coordinates.  A very well known result is that for a particle traveling with a velocity less than c, there is no possibility to cross the horizon and then go out. But this is possible if the velocity is higher than c. A classical tachyon has precisely this property, so it is not unexpected that it will escape from the BH horizon. Am I missing something?

4.  The open questions remarked by the author in section 9 are certainly important to be addressed.  Although I understand that at this moment there are no answers for those questions, I would like to know how the tachyonic nature of particles could add to the understanding of such phenomena.  What is the idea?

Therefore, I do not accept the paper to be published in the form it is. Mayor changes are required, mainly involving adding references on this topic by other researchers and by mentioning how the proposal of tachyonic matter can replace the Dark Matter hypothesis beyond the stellar orbits.

  1.  
  2.  

Round 2

Reviewer 2 Report

The author has addressed all my comments. In virtue of his response, I accept the manuscript in present form  to be published in this journal.